



# Chemical composition and droplet size distribution of cloud at the summit of Mount Tai, China

Jiarong Li[1], Xinfeng Wang[1], Jianmin Chen[1,2,3,*], Chao Zhu[1], Weijun Li[1], Chengbao Li[2], Lu Liu[1], Caihong Xu[1], Liang Wen[1], Likun Xue[1], Wenxing Wang[1], Aijun Ding[3], Hartmut Herrmann[2,4,*]

[1] Environment Research Institute, School of Environmental Science and Engineering, Shandong University, Ji'nan 250100, China.

[2] Shanghai Key Laboratory of Atmospheric Particle Pollution and Prevention, Department of Environmental Science and Engineering, Institute of Atmospheric Sciences, Fudan University, Shanghai 200433, China.

[3] Institute for Climate and Global Change Research, School of Atmospheric Sciences, Nanjing University, Nanjing 210023, Jiangsu, China

[4] Leibniz Institute for Tropospheric Research (TROPOS), Atmospheric Chemistry Department (ACD), Permoserstr. 15, D-04318, Leipzig, Germany.

*Correspondence to*: J. M. Chen (jmchen@sdu.edu.cn, jmchen@fudan.edu.cn); H. Herrmann (herrmann@tropos.de)

**Abstract.** Chemical composition of 39 cloud samples and droplet size distribution in 24 cloud events were investigated at the summit of Mt. Tai from July to October 2014. Inorganic ions, organic acids, metals, HCHO, $H_2O_2$, sulfur(IV), organic carbon, element carbon as well as pH and electrical conductivity were analyzed. The acidity of the cloud water significantly decreased from a reported value of pH 3.86 in 2007–2008 (Guo et al., 2012) to pH 5.87 in the present study. The concentrations of nitrate and ammonium were both increased since 2007-2008, but the overcompensation of ammonium led to the increase of the mean pH value. The microphysical properties showed that cloud droplets were smaller than 26.0 μm and the most were in the range of 6.0–9.0 μm. The maximum droplet number concentration ($N_d$) was associated with droplet sizes of 7.0 μm. Cloud droplets exhibited a strong interaction with atmospheric aerosols. High $PM_{2.5}$ level resulted in higher concentrations of water soluble ions and smaller sizes with more numbers of cloud droplets, and further gave rise to relatively high acidity. High degrees of relative humidity facilitated the formation of large cloud droplets and led to high liquid water contents under low $PM_{2.5}$ level. The cloud droplets to wet deposition acted as an important sink of soluble material in the atmosphere and the dilution effect of the water content should be considered when estimating concentrations of soluble components in the cloud phase.

Keywords: Chemical compositions, Cloud droplet size distribution, Cloud scavenging, Mount Tai.

## 1 INTRODUCTION

Cloud droplets are formed by the condensation of water vapor on anthropogenic and natural aerosols that serve as cloud





condensation nuclei (CCN). Clouds significantly affect the earth's radiation budget and they are also responsible for changes in regional and global climate (Miles et al., 2000). Cloud events can transport pollutants, promote acid deposition, change the meteorological conditions, modify local environmental features and affect the fate of several atmospheric species via chemical and physical processes (Moore et al., 2004).

The chemical properties of clouds are initially determined by the CCN (Sun et al., 2010) but they can be altered as a result of absorbing chemical components of soluble gases and further taking place multiphase chemical reactions (Ravishankara, 1997). Non-precipitating clouds play a more crucial role in ion deposition and aggregation than precipitating clouds (Aleksic et al., 2009). The concentrations of soluble compounds and dissolved acids have generally been reported to be much higher in cloud liquid water compared with precipitation (Błaś et al., 2008; Zapletal et al., 2007; Zimmermann et al., 2003). For example,

Shimadera and colleagues found in certain regions with orographic clouds, more than 30% of the total annual sulfur deposition was deposited as a result of cloud events (Shimadera et al., 2011).

    Cloud plays a significant role in scavenging aerosols via drop deposition (directly or by coalescence into precipitation) and in creating new particles and trace gases (Herckes et al., 2002). These processes influence the distribution and concentration of pollutants in both the cloud phase and the aerosol phase, and they also influence the microphysical properties of the clouds

(Collett Jr et al., 2002; Lee et al., 2012; Ogawa et al., 2000). For example, for a given supersaturated condition, an increase in the concentration of CCN will lead to the formation of small droplets (Borys et al., 2000; Gultepe and Milbrandt, 2007). In addition, the cloud droplet size distribution (CDSD) is prominently determined by the chemical and physical properties of the CCN (Portin et al., 2013; Zipori et al., 2015). Numerous studies have examined the chemical composition of orographic clouds (Kim et al., 2006; Marinoni et al., 2004; Watanabe et al., 2010), many of which have focused on the size-dependent chemical

properties of the clouds (Moore et al., 2004; Schell et al., 1997). However, few studied provide detailed descriptions of the interactions between aerosols and the chemical and microphysical properties of clouds.

    In this study, cloud samples were collected at the summit of Mount Tai. It is interesting that the acidity of the cloud water was significantly lower than that reported in 2007–2008 (Guo et al., 2012; Wang et al., 2011). The causes behind this change were investigated by examining the changes in the chemical compositions of the cloud samples. We then investigated the

microphysical properties of cloud droplets, including cloud droplet size distribution (CDSD), liquid water content (LWC), and droplet number concentration ($N_d$). Lastly, we explored the interactions between cloud droplets and the atmospheric aerosols.

## 2 METHODS

### 2.1 Site description and sampling

Mount Tai (117°13′E, 36°18′N, 1545 m a.s.l.) is a natural and cultural heritage site in China and one of the world's geoparks.

The summit of Mt. Tai is lack of emissions of anthropogenic pollutions, so the pollutants in cloud samples collected at the summit could accurately represent the characteristics of the regional pollutants in the North China Plain. The local high



frequency of cloud events, especially in summer, makes Mount Tai a favorable site for collecting cloud samples and monitoring cloud events. Previous research has indicated that the clouds at the summit of Mount Tai are acidic (Wang et al., 2008).

From July 24 to October 31, 2014, a total of 85 cloud samples associated with 24 cloud events were collected using a single–stage Caltech Active Strand Cloud Water Collector (CASSC), as described by (Demoz et al., 1996) and 39 cloud samples were

analyzed. The cloud droplets were inhaled into the collector by a fan with a flow rate of 24.5 $m^3$ $min^{-1}$ and impacted on six Teflon nets that each contained 102 strands of 508 μm in diameter. The samples were then guided along a groove at the bottom of the collector and finally collected into a 500 mL high-density polyethylene cylinder. The theoretical 50% collection efficiency cut size of the cloud droplets is at 3.5 μm. In this study, sampling time resolution was adjusted during sampling sessions in order to ensure that each sample contained an adequate amount of cloud water (at least 150 mL) for the analysis.

The volumes of the samples, the start and end times of the collection sessions and the numbers of collected samples were accurately recorded for each cloud event.

It should be noted that the collector was immediately shut down during precipitation to eliminate the interruptions caused by rain water. Before each sampling session, the collector was rinsed with high-purity deionized water ($\geq$ 18.2 MΩ), dried naturally and sealed. Blanks were prepared using high-purity deionized water, and then they were treated and analyzed using

the same method as collected samples.

## 2.2 In-situ and laboratory analysis

The pH, the electrical conductivity and the concentrations of sulfur(IV), formaldehyde and hydrogen peroxide were measured immediately after sampling. Approximately 10 mL of each cloud sample was used to measure the pH and electrical conductivity using a portable pH meter (model 6350M, JENCO) that was regularly calibrated using standard solutions at pH

=4 and pH =7. Approximately 20 mL of each cloud sample was filtered using a cellulose acetate filter with pore sizes of 0.45 μm to remove any suspended particulate matter and then the concentrations of sulfur(IV), formaldehyde and hydrogen peroxide were analyzed in-situ to avoid any changes in their concentrations. The measurement methods were described in detail by Collett and colleagues (Collett Jr et al., 1998). For each sample, a 10 mL aliquot was prepared for trace metal analysis by adding 1% (v/v) nitric acid and then preserved in a brown glass bottle at 4 ℃. Another 10 mL aliquot was prepared to analyze

organic acids by adding 0.5% (v/v) chloroform (to prevent the reproduction of microorganisms) and then storing the solution in a glass bottle at 4 ℃. The residuals were refrigerated at -20 ℃ until further analyzing.

The concentrations of eight inorganic ions ($Cl^-$, $NO_3^-$, $SO_4^{2-}$, $NH_4^+$, $Na^+$, $K^+$, $Ca^{2+}$ and $Mg^{2+}$) in each sample were measured using ion-chromatography (Dionex, ICS-90) and the concentrations of four organic acids (acetate, formate, oxalate and lactate) were measured using ion-chromatography (Dionex, IC-2500). Trace metals such as Fe and Mn were analyzed using inductively

coupled plasma mass spectrometry (ICP-MS; Agilent 7500a). The concentrations of organic carbon (OC) and elemental carbon (EC) in each sample were measured using an OC/EC analyzer (Sunset Lab).



## 2.3 Monitoring of microphysical parameters

A fog monitor (model FM-120, Droplet Measurement Technologies Inc., USA) was used in-situ to monitor the liquid water content (LWC), the median volume diameter (MVD), the effective diameter (ED) and the droplet number concentration ($N_d$) of the cloud droplets with a time resolution of 1 s. During July 24 to August 23, 2014, 24 cloud events were monitored. The measuring range of cloud droplets diameter is from 2–50 μm in 20 bins. The sample velocity is 15 m s$^{-1}$ and the sampling flow is 1 m$^3$ min$^{-1}$. Cloud droplets cannot be collected efficiently at low LWC and $N_d$ values. Based on our experience, the sampling limitations associated with LWC and $N_d$ were 0.01 g m$^{-3}$ and 60 # cm$^{-3}$, respectively.

## 2.4 Measurements of ambient air pollutants and meteorological parameters

The concentrations of inorganic water–soluble ions, the levels of PM$_{2.5}$ and the meteorological parameters were monitored in real-time during the observation periods. The $SO_4^{2-}$, $NO_3^-$ and $NH_4^+$ in PM$_{2.5}$ were measured using two on-line ion chromatographs coupled with a wet rotating denuder and a steam–jet aerosol collector (MARGA ADI 2080, Applikon-ECN). A Beta attenuation and optical analyzer (model 5030 SHARP monitor, Thermo Scientific) was used to monitor the levels of PM$_{2.5}$. Meteorological parameters including the ambient temperature, relative humidity, wind speed and wind direction were measured using an automatic meteorological station.

## 3 RESULTS AND DISCUSSION

### 3.1 Chemical properties of cloud water

### 3.1.1 Acidity

The pH values, the electrical conductivity and chemical compositions of the cloud droplets (inorganic ions, organic acids, metals, HCHO, $H_2O_2$, sulfur(IV), OC, and EC are summarized in Table 1. The pH of the cloud water varied widely from 3.80–6.93. The volume-weighted mean (VWM) pH was 5.87, which is slightly higher than the background pH of 5.6 yielded by $CO_2$ in the atmosphere. The analyzed 39 cloud samples were divided into one group of 17 summer samples (i.e., those that were collected from July to August) and a second group of 22 autumn samples (i.e., those that were collected from September to October). About 52% of the summer samples was under pH of 5.6 and 12% were under pH of 4.5. The corresponding percentages for the autumn samples were 14% and 9%, respectively. We found that some of the cloud samples were acidic, especially in the summer. If compared with other orographic stations less affected by anthropogenic pollutions, the VWM pH of clouds at Mount Tai was higher as shown in Table 2. Moreover, the VWM pH at Mount Tai significantly increased from a reported value of 3.86 in 2007–2008 (Guo et al., 2012) to 5.87 in the present study. The detailed reasons for the big decrease in cloud water acidity are discussed in the later section.



### 3.1.2 Ion composition

The cloud samples contained high concentrations of water-soluble ions. The dominant ions were nitrate, sulfate, ammonium and calcium by the VWM concentrations of 56.4, 44.2, 34.2 and 5.9 mg L$^{-1}$, respectively. These ions represented 88.1% of the total determined ion concentrations (TDIC). The concentrations of minor ions including chloride, potassium, sodium, magnesium and organic acids ranged from 0.7 mg L$^{-1}$ to 4.1 mg L$^{-1}$ and amounted to only 10.6% of the TDIC. As a result of the high levels of agricultural and livestock activities taking place near Mount Tai, $NH_4^+$ was the predominant cation (Cai et al., 2015; Xu et al., 2015). Calcium was the second most abundant cation and was likely to have originated from sandstorms and/or construction activities. The concentration of $SO_4^{2-}$ amounted to 27.7% of the TDIC, which made $SO_4^{2-}$ the second most abundant anion. The concentration of non-sea salt sulfate (nss-$SO_4^{2-}$) was calculated using the equation [nss-$SO_4^{2-}$] = [$SO_4^{2-}$]-0.2455[$Na^+$]. In this calculation, it was assumed that the chemical properties of sea salt sulfate (ss-$SO_4^{2-}$) in particles are identical to those in sea water and that the soluble $Na^+$ originated solely from sea salt (Morales et al., 1998). The nss-$SO_4^{2-}$ represented 93.5–100% of the total $SO_4^{2-}$ and might have been underestimated as, besides sea salts, soil dust and biomass combustion are also sources of $Na^+$ (Lu et al., 2010; Sripa et al., 1996). The high ratio of ss-$SO_4^{2-}$ to $SO_4^{2-}$ indicated that anthropogenic sulfur emissions were the main sources of $SO_4^{2-}$ in the cloud samples from Mount Tai. It should be noted that the VWM of the concentration of $SO_4^{2-}$ was almost the same as that reported in 2007–2008, but the concentration of $NO_3^-$ increased significantly by a factor of 2.24 (Guo et al., 2012). This made $NO_3^-$ surpass $SO_4^{2-}$ and be the predominant anion in 2014. Generally, the scavenging of aerosol nitrate and the uptake of gaseous nitric acid are the main sources of nitrate in cloud/fog water (Collett Jr et al., 2002). It implies that there has been a substantial increase in nitrate precursor emission, which are likely to have involved $NO_x$ from power plants and motor vehicles.

Generally, the pH of cloud water is determined by the balance between the acid and the alkaline components. Two factors can decrease the acidity of cloud water: a large input of alkaline ions and/or a decrease in acid anions. Although the VWM concentration of $NO_3^-$ increased significantly, the additional increases in $NH_4^+$ and $Ca^{2+}$ should also be noted. The VWM concentrations of $NH_4^+$ and $Ca^{2+}$ increased from 2007–2008 by factors of 1.56 and 1.53, respectively (Guo et al., 2012), which may be attributable to the increasing consumption of agricultural fertilization and soil acidification (Cai et al., 2015; Xu et al., 2015). As a result, the increased levels of $NH_4^+$ and $Ca^{2+}$ played a crucial role in neutralizing the soluble acid ions ($NO_3^-$ and $SO_4^{2-}$) and decreased the acidity of cloud water since 2007–2008.

### 3.2 Microphysical properties of cloud water

### 3.2.1 Microphysical parameters

The sampling period, number of cloud samples, mean level of PM$_{2.5}$, mean microphysical parameters and meteorological conditions for each cloud event are summarized in Table 3. There was a great deal of diversity in the $N_d$ and the LWC among the cloud events. The mean values of $N_d$ ranged widely from 79 # cm$^{-3}$ to 722 # cm$^{-3}$ and the mean values of LWC ranged widely from 0.01 g m$^{-3}$ to 0.39 g m$^{-3}$. This diversity was a result of the characteristic formation of the orographic clouds, which





generally determines the differences in CDSD, LWC, aerosol number and chemical composition (Gonser et al., 2012).

The microphysical properties of the cloud droplets were related to the $PM_{2.5}$ levels. High $PM_{2.5}$ levels can lead to low LWC values, which can diminish the size of the cloud droplets. As can be seen, the mean $PM_{2.5}$ levels of cloud events 3, 4, 12, 15 and 17 were all greater than 75.0 µg m$^{-3}$, leading to low LWC values (lower than 0.10 g m$^{-3}$) and small cloud droplets (the ED values were lower than 7.8 µm). However, in events 21 and 24, the levels of $PM_{2.5}$ determined neither the LWC values nor the ED values. In these two events, the levels of $PM_{2.5}$ were not very high (57.9 and 29.4 µg m$^{-3}$, respectively), but the LWC values were very low (0.02 and 0.01 g m$^{-3}$, respectively) and the cloud droplets were smaller than 6.5 µm. This was due to the low relative humidity (RH), which did not supply sufficient water vapor to promote the growth of the cloud droplets. In summary, high levels of $PM_{2.5}$ can lead to a large source of CCN and intensify the competition for the ambient water vapor. If the RH remains constant, each CCN shares less water vapor, which leads to lower LWC values and hinders the growth of cloud droplets. If RH varies, the size of cloud droplets is determined by the combined effect of the RH and the $PM_{2.5}$ level.

### 3.2.2 Cloud droplet size distribution

The cloud droplet size distribution, which indicates the dynamic and thermodynamic properties of a cloud system, is one of the most crucial determinants of the microstructures of cloud (Yin et al., 2011). To investigate the CDSD, four typical cloud events (A, B, C and D) were studied in light of their mean $PM_{2.5}$ levels of 81.6 (A), 43.0 (B), 25.0 (C) and 11.1 µg m$^{-3}$ (D), respectively. As shown in Fig. 1, all of the cloud droplets in cloud samples were smaller than 26.0 µm. As the cloud processes continued, droplets ranging from 6.0–9.0 µm became dominant. The ratio of cloud droplets with 6.0–9.0 µm to all droplet sizes was relatively stable among the four cloud events (i.e., between 0.6–0.7: 1). The maximum $N_d$, which reached over 1950 # m$^{-3}$, always occurred at a droplet size of 7.0 µm.

An examination of the meteorological parameters with the microphysical properties of the clouds showed that the ambient temperature and the LWC somewhat influenced the CDSD. Higher temperatures and higher LWC values increased the numbers of larger cloud droplets and broadened the droplet size spectra, while lower temperatures and lower LWC values inhibited the formation of larger cloud droplets. The formation stage of cloud event B, which occurred at 1:30–2:40 on August 23, 2014, provided a clear example. During this event, both the temperature and the LWC were relatively low with mean values of 16.7 ℃ and 0.09 g m$^{-3}$, respectively. At 2:30, about 8.6% of the cloud droplets had diameters above 10.0 µm and 27.6% had diameters below 5.0 µm. After 8 min, the corresponding values were 16.3% and 17.1%, respectively, as the temperature increased to 17 ℃ and the LWC sharply increased to 0.29 g m$^{-3}$. Moreover, cloud droplets that were larger than 16.0 µm started to appear and the CDSD changed from a monomodal distribution to a weakly bimodal distribution. This situation also occurred in many other cloud events at Mount Tai. It should be emphasized that although the levels of $PM_{2.5}$ decreased from event A to event D, there were no significant changes in the CDSD properties. This suggests that other factors (rather than the $PM_{2.5}$ level) have more influence on the CDSD.




### 3.2.3 Cloud scavenging effect

Cloud processes together with wet deposition play crucial roles in scavenging atmospheric aerosols. Based on the initial $PM_{2.5}$ levels, cloud processes can be classified into two types: type I (including events A and B) that have high initial $PM_{2.5}$ levels and type II (including events C and D) that have low initial $PM_{2.5}$ levels.

Type I cloud processes existed high levels of aerosol scavenging activity. Using event A as an example, at the beginning of the cloud process, there was a relatively high level of $PM_{2.5}$ (approximately 128 μg m$^{-3}$) and $N_d$ increased sharply from 6 # cm$^{-3}$ to 437 # cm$^{-3}$ over 1 min. As the cloud process continued, the level of $PM_{2.5}$ decreased and then fluctuated with a mean concentration of 78.2 μg m$^{-3}$. About 30 min later, the $N_d$ reached the maximum with 1538 # cm$^{-3}$ and the level of $PM_{2.5}$ reached the minimum with 23.9 μg m$^{-3}$, which indicated a high $PM_{2.5}$ removal efficiency of 81.3%. The somewhat inverse relationship

between $N_d$ and the level of $PM_{2.5}$ reflects the efficient pollutant removal effect of cloud formation. As the cloud process continued, the cloud began to dissipate and the $N_d$ decreased significantly. At the same time, the level of $PM_{2.5}$ evidently increased and reached a new peak about 104.5 μg m$^{-3}$. This may due to the evaporation of water contents that condensed on the aerosols, which freed the CCN and formed haze. These results suggest that aerosols that act as CCN can be efficiently cleared during (rather than after) cloud processes.

In the type II cloud processes, the initial levels of $PM_{2.5}$ were relatively constant but increased sharply as the cloud events came to an end. It should be noted that the air mass involved in events A and B were primarily from the west of China and they were clearly influenced by the transportation of dust from sandstorms (as indicated by the high concentrations of $Ca^{2+}$ and $PM_{2.5}$ in the cloud samples from these two events, which are shown in Fig. 1). However, the wind in events C and D primarily came from the east and southeast of China and travelled through a dense urban region. Thus, we inferred that

municipal pollution may be the important factor induced the increase of $PM_{2.5}$ at the dissipation stage of type II cloud processes.

### 3.3 Interaction between aerosols and cloud chemical properties

As illustrated in Fig. 2, the TDIC was strongly correlated with the levels of $PM_{2.5}$ and cloud acidity. High levels of $PM_{2.5}$ normally lead to high TDIC and low pH values, whereas low levels of $PM_{2.5}$ usually lead to low TDIC and high pH values. Generally, changes in the solute concentrations of cloud water can be caused by a combination of factors such as the

25 microphysical conditions, the CCN properties, the chemical reactions in the cloud droplets and the gas-liquid phase equilibrium (Van Pinxteren et al., 2015). Our data emphasized the crucial effect of CCN on changes of ion concentrations and cloud acidity. CCN, especially particulate matters, are likely to be the main source of ions and acid-causing components in cloud water.

     To understand the transmission and variation of the three major ions ($SO_4^{2-}$, $NO_3^-$ and $NH_4^+$) between the aerosol phase and the cloud phase at the summit of Mount Tai, we analyzed three cloud samples (CE-Aug23#1 from 02:30–04:38, CE-Aug23#2

from 04:38–06:21 and CE-Aug23#3 from 06:21–09:20) that were collected from the same cloud event (event B on Aug. 23, 2014). As shown in Fig. 3, in the aerosol phase, the concentrations of $SO_4^{2-}$, $NO_3^-$ and $NH_4^+$ decreased with increases of LWC and vice versa. In the cloud phase, high LWC values meant large cloud droplets and low concentrations of major ions while



low LWC values induced small cloud droplets with high levels of $SO_4^{2-}$, $NO_3^-$ and $NH_4^+$. Elbert and colleagues also observed an inverse relationship between the ion concentrations and the LWC values (Elbert et al., 2000). Between CE-Aug23#1 and CE-Aug23#2, the ion concentrations decreased by factors of 2.29, 2.07 and 1.51 for $SO_4^{2-}$, $NO_3^-$ and $NH_4^+$, respectively. Meanwhile, the LWC increased from 0.04 g m$^{-3}$ to 0.32 g m$^{-3}$ and ED increased from 6.7 μm to 10.2 μm. At the dissipation stage of the cloud event, the LWC decreased to less than 0.10 g m$^{-3}$ and ED shrank to about 6.6 μm. Simultaneously, the ion concentrations significantly increased by factors of 1.18, 1.56 and 1.40 for $SO_4^{2-}$, $NO_3^-$ and $NH_4^+$, respectively.

The above results demonstrate that cloud water is a dominant sink of soluble ions in the atmosphere and small cloud droplets tend to contain high concentrations of soluble ions than larger ones. $SO_4^{2-}$, $NO_3^-$ and $NH_4^+$ in the aerosol phase were primarily assumed to be transferred to the cloud phase. However, the concentrations of the soluble components in the cloud phase could not be accurately predicted based only on their concentrations in the aerosol phase, as the strong dilution effect of the cloud water content must also be considered. The concentrations of ions in the cloud phase were primarily determined by two factors: the sources of the ions (i.e., the corresponding ion concentrations in the aerosols acted as CCN) and the LWC values (which represents the dilution effect of the cloud water). The similar variation trends of $SO_4^{2-}$, $NO_3^-$ and $NH_4^+$ in both aerosol phase and cloud phase confirmed that the LWC values rather than aerosols was a more important determinant of ion concentrations in the cloud water at Mt. Tai. As mentioned above, LWC also determined the size of cloud droplets. This ultimately represented that high concentrations of soluble ions concentrated in small cloud droplets. It should be noted that the increase in the concentration of $NH_4^+$ from CE-Aug23#2 to CE-Aug23#3 was much higher than those of $SO_4^{2-}$ and $NO_3^-$, which delayed the rise of $NH_4^+$ concentration in the aerosol phase. This was primarily due to the dissolution of atmospheric $NH_3$ in the cloud water.

## 3.4 Water soluble ions and droplet size under PM$_{2.5}$

Secondary inorganic aerosols especially ammonium sulfate and ammonium nitrate were the main hygroscopic compounds. The hygroscopic behaviour of these atmospheric aerosols may facilitate their ability to act as cloud condensation nuclei (Wang et al., 2014; Ye et al., 2011). These water soluble ions are primarily transferred to the cloud phase during the formation of cloud droplets by activation of aerosol CCN. As mentioned before, $SO_4^{2-}$, $NO_3^-$, $NH_4^+$ and $Ca^{2+}$ were the most predominant ions in cloud samples collected at Mt. Tai. The averaged concentrations surpassed 88.1% of the total determined ion concentrations. Presumably, PM$_{2.5}$ was the main source of the mentioned soluble ions in cloud water. In order to investigate the variation trend between water soluble ions and cloud droplet size under different PM$_{2.5}$ levels, 17 cloud samples collected from 25 July to 23 August were studied as shown in Fig. 4. As can be seen, high PM$_{2.5}$ level represented high ion concentrations and small cloud droplets. It confirmed again that PM$_{2.5}$ acted as CCN was the main source of soluble ions in cloud water. High PM$_{2.5}$ levels would increase the competition of ambient water vapor and hinder the formation of large cloud droplets.

It should be noticed that sometimes the $N_d$ varied with the same PM$_{2.5}$ level in Fig. 4b. It was caused by the variation of LWC values in different monitoring moments. Even though PM$_{2.5}$ level was high, low water content in the atmosphere could not provide enough water for the formation of cloud droplets.



**4 Conclusions**

Samples of clouds showed that the VWM pH of the cloud samples was 5.87, which is much higher than that reported by previous studies that took place at the same site in 2007–2008. The cloud water contained much higher concentrations of ions than the samples collected at other orographic sites, indicating the strong influence of anthropogenic emissions on clouds at

the summit of Mount Tai. The dominant ion species were $NH_4^+$, $SO_4^{2-}$, $Ca^{2+}$ and $NO_3^-$, which amounted to more than 88.1% of the TDIC. The $NO_3^-$ content of the cloud water was significantly higher than that in 2007–2008. However, the increase of the $NH_4^+$ concentration (mainly from $NH_3$) exceeded that of $NO_3^-$ (mainly from $NO_x$), leading to net neutralization and reduced the cloud acidity. The rapid increases in the concentrations of $NH_4^+$ and $Ca^{2+}$ are attributable to the increases in agricultural fertilization and soil acidification that have occurred in recent years (Cai et al., 2015; Xu et al., 2015). The microphysical

parameters of the cloud samples varied enormously between the cloud events. The cloud droplets were all smaller than 26.0 μm and most were 6.0–9.0 μm. The maximum $N_d$ was associated with droplet sizes of 7.0 μm. High RH and low $PM_{2.5}$ levels facilitated the growth of cloud droplets, which in turn increased the LWC. High temperatures and high LWC slightly increased the number of large cloud droplets and broadened the droplet size spectra. A strong interaction was observed between the cloud droplets and the atmospheric aerosols. The clouds played a crucial role in scavenging atmospheric aerosols. Higher $PM_{2.5}$ level

resulted in higher TDIC, which lowered the pH of the cloud samples. We found that the dilution effect of cloud water was strong and it should not be ignored when estimating concentrations of soluble components in the cloud phase.

In summary, the mechanism of cloud droplet formation is summarized in Fig. 5. Cloud droplets would be formed on condensation nuclei (usually aerosols including secondary aerosol, dust, sea salt, and so on) through water vapor condensation and then undergo hygroscopic growth. The soluble ions in condensation nuclei and ambient gases could enter cloud droplets

through surface reactions and consequently participate dissolution, diffusion, dilution and aqueous reaction in the cloud phase. Higher aerosol concentrations supplied higher concentrations of soluble ions for cloud droplets and facilitate the formation of smaller sizes of cloud droplets, which caused the high concentrations of soluble ions in small cloud droplets.

**ACKNOWLEDGEMENTS**

This work was supported by Taishan Scholar Grant (ts20120552), National Natural Science Foundation of China (41375126,

41275123, 21190053, 21177025), Cyrus Tang Foundation (No. CTF-FD2014001), Ministry of Science and Technology of China (2016YFC0202701, 2014BAC22B01), Strategic Priority Research Program of the Chinese Academy of Sciences (Grant No. XDB05010200), Natural Science Foundation of Shandong Province (No. ZR2014BQ031) and Marie Skłodowska-Curie Actions (H2020-MSCA-RISE-2015-690958).

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





**List of Table and Figure Captions**

Table 1. Summary of the chemical compositions for cloud samples collected at Mt. Tai during July to October, 2014.

Table 2. Comparison of averaged ionic concentrations ($\mu$eq L$^{-1}$) of cloud water collected at Mt. Tai with other regions in the world.

Table 3. Description of monitored cloud events at Mt. Tai with monitoring times, number of samples (No. Samples) and averaged values of liquid water content (LWC), median volume diameter (MVD), effective diameter (ED), number concentration ($N_d$), temperature (T) and relative humidity (RH).

Figure 1: The variation of wind speed (m s$^{-1}$), wind direction, T (Temperature, ℃), RH (Relative Humidity, %), PM$_{2.5}$ level ($\mu$g cm$^{-3}$), LWC (Liquid Water Content, g m$^{-3}$) and Cloud Droplet Number Concentration (CDNC) during four typical cloud

events: event A (28/07/2014 22:40 to 29/07/2014 04:00); event B (23/08/2014 01:30 to 23/08/2014 09:20); event C (30/07/2014 20:20 to 30/07/2014 22:40) and event D (25/07/2014 12:00 to 25/07/2014 21:40).

Figure 2: Ion and organic acid concentrations ($\mu$eq L$^{-1}$) with the variation of PM$_{2.5}$ levels ($\mu$g·cm$^{-3}$) and pH of cloud water samples.

Figure 3: Variation trend of hour averaged LWC (g m$^{-3}$), ED ($\mu$m) and the concentrations of NO$_3^-$, SO$_4^{2-}$ and NH$_4^+$ in aerosol

phase and cloud phase during the cloud event on August 23, 2014.

Figure 4: (a) The variation of ED of cloud droplets and the sum of four water soluble ions (SO$_4^{2-}$, NO$_3^-$, NH$_4^+$ and Ca$^{2+}$) under different PM$_{2.5}$ levels (b) The variation of ED and $N_d$ of cloud droplets under different PM$_{2.5}$ levels.

Figure 5: The schematic of aerosol particles' impact on the cloud droplet sizes. a: PM$_{2.5} \geq$ 40 $\mu$g m$^{-3}$, b: PM$_{2.5} <$ 40 $\mu$g m$^{-3}$.





**Table 1.**

| Species | Units | No. Samples | Min | Max | VWM[b] | Percentage |
|---|---|---|---|---|---|---|
| pH | --- | 39 | 3.80 | 6.93 | 5.87 | --- |
| Electrical Conductivity | $\mu S\ cm^{-1}$ | 39 | 44.9 | 813.5 | 169.0 | --- |
| $Na^+$ | $mg\ L^{-1}$ | 39 | BDL[a] | 2.9 | 0.9 | 0.56 |
| $NH_4^+$ | $mg\ L^{-1}$ | 39 | 5.2 | 143.3 | 34.2 | 21.41 |
| $K^+$ | $mg\ L^{-1}$ | 39 | BDL[a] | 6.5 | 1.3 | 0.81 |
| $Mg^{2+}$ | $mg\ L^{-1}$ | 39 | 0.2 | 3.0 | 0.7 | 0.44 |
| $Ca^{2+}$ | $mg\ L^{-1}$ | 39 | BDL[a] | 39.2 | 5.9 | 3.69 |
| $Cl^-$ | $mg\ L^{-1}$ | 39 | 0.6 | 11.7 | 2.9 | 1.82 |
| $NO_3^-$ | $mg\ L^{-1}$ | 39 | 2.7 | 538.5 | 56.4 | 35.31 |
| $SO_4^{2-}$ | $mg\ L^{-1}$ | 39 | 10.5 | 253.0 | 44.2 | 27.67 |
| $nss\text{-}SO_4^{2-}$ | $mg\ L^{-1}$ | 39 | 10.5 | 251.6 | 43.7 | --- |
| lactate | $mg\ L^{-1}$ | 13 | BDL[a] | 7.8 | 3.0 | 1.88 |
| acetate | $mg\ L^{-1}$ | 15 | BDL[a] | 14.9 | 4.1 | 2.57 |
| formate | $mg\ L^{-1}$ | 17 | 0.4 | 14.4 | 2.8 | 1.75 |
| oxalate | $mg\ L^{-1}$ | 17 | 0.6 | 3.6 | 1.3 | 0.81 |
| Mn | $mg\ L^{-1}$ | 39 | 0.01 | 0.28 | 0.04 | 0.03 |
| Fe | $mg\ L^{-1}$ | 39 | 0.06 | 3.02 | 0.40 | 0.25 |
| HCHO | $mg\ L^{-1}$ | 39 | BDL[a] | 5.9 | 0.4 | 0.25 |
| $H_2O_2$ | $mg\ L^{-1}$ | 39 | BDL[a] | 3.3 | 0.8 | 0.50 |
| S(IV) | $mg\ L^{-1}$ | 39 | BDL[a] | 1.1 | 0.4 | 0.25 |
| $OC^c$ | $mg\ L^{-1}$ | 17 | BDL[a] | 211.8 | 37.4 | --- |
| $EC^d$ | $mg\ L^{-1}$ | 17 | BDL[a] | 8.5 | 0.3 | --- |
| Average $PM_{2.5}$ Level | $\mu g\ m^{-3}$ | 39 | 0.7 | 81.6 | 15.9 | --- |

[a] BDL means Below Detection Limit
[b] Volume Weighted Mean Concentration
[c] OC means Organic Carbon
[d] EC means Element Carbon





**Table 2.**

| Site | Period | Altitude (m) | pH | EC (µS cm$^{-1}$) | Na$^+$ | NH$_4^+$ | K$^+$ | Mg$^{2+}$ | Ca$^{2+}$ | Cl$^-$ | NO$_3^-$ | SO$_4^{2-}$ | Reference |
|---|---|---|---|---|---|---|---|---|---|---|---|---|---|
| Whiteface Mountain, NY USA | May-Sept 2006 | 1483 | 3.88 | 79.6 | 3.7 | 149.3 | 2.1 | 7.4 | 26.6 | 7.2 | 79.2 | 220.4 | (Aleksic et al., 2009) |
| Szrenica, Poland | Dec 2005-Dec 2006 | 1330 | 4.55 | 80 | 100 | 210 | 45 | 49 | 140 | 93 | 240 | 200 | (Błaś et al., 2010) |
| Mt. Niesen, Swizerland | 2006-2007 | 2362 | 6.4 | 34.4 | 43 | 143.5 | 5 | 12.6 | 46.8 | 10.6 | 87 | 72.3 | (Michna et al., 2015) |
| Sinhagad, India | 2007-2010 | 1450 | 6 | 86 | 204 | 28 | 17 | 17 | 196 | 234 | 68 | 198 | (Budhavant et al., 2014) |
| Mt. Heng, China | Mar-May 2009 | 1279 | 3.8 | 115.26 | 66.03 | 356.47 | 17.25 | 5.49 | 29.83 | 21.07 | 158.8 | 196.39 | (Sun et al., 2010) |
| Mt. Tai, China | 2007-2008 | 1545 | 3.86 | --- | 25.0 | 1215 | 55.1 | 33.0 | 193 | 93.4 | 407 | 1064 | (Guo et al., 2012) |
| Mt. Tai, China This Work | Jul-Oct 2014 | 1545 | 5.87 | 169.0 | 39.7 | 1900.8 | 32.7 | 60.5 | 295.5 | 82.5 | 910.2 | 920.9 | |



**Table 3.**

| Event Number | Start (UTC/GMT+8) | Stop (UTC/GMT+8) | No. Samples | Duration (h) | [a]$PM_{2.5}$ ($\mu g\ m^{-3}$) | [a]LWC ($g\ m^{-3}$) | [a]$N_d$ (# $cm^{-3}$) | [a]MVD ($\mu m$) | [a]ED ($\mu m$) | T (°C) | [a]RH (%) |
|---|---|---|---|---|---|---|---|---|---|---|---|
| 1 | 24/07/2014 08:30 | 24/07/2014 23:20 | 3 | 14.8 | 14.5 | 0.24 | 408 | 12.7 | 11.0 | 15.5-22.6 | 97.9 |
| 2 | [b]25/07/2014 12:00 | 25/07/2014 21:40 | 2 | 9.7 | 11.1 | 0.18 | 719 | 8.3 | 8.3 | 13.6-14.6 | 100.0 |
| 3 | 26/07/2014 23:06 | 27/07/2014 05:13 | 0 | 6.1 | 100.7 | 0.04 | 211 | 7.8 | 7.4 | 15.7-17.0 | 99.0 |
| 4 | [b]28/07/2014 22:40 | 29/07/2014 04:00 | 1 | 5.3 | 81.6 | 0.09 | 337 | 8.2 | 7.8 | 16.5-17.6 | 99.2 |
| 5 | 29/07/2014 20:33 | 29/07/2014 22:20 | 0 | 1.8 | 65.6 | 0.14 | 694 | 7.8 | 7.6 | 18.5-18.9 | 99.3 |
| 6 | 30/07/2014 12:46 | 30/07/2014 13:50 | 1 | 1.1 | 13.2 | 0.21 | 308 | 12.6 | 11.8 | 16.8-18.5 | 99.5 |
| 7 | [b]30/07/2014 20:20 | 30/07/2014 22:40 | 0 | 2.3 | 25.0 | 0.08 | 253 | 9.2 | 9.2 | 16.9-18.2 | 99.6 |
| 8 | 31/07/2014 19:11 | 01/08/2014 09:19 | 2 | 14.1 | 20.1 | 0.18 | 329 | 12.6 | 11.5 | 17.9-19.1 | 99.5 |
| 9 | 04/08/2014 23:42 | 05/08/2014 11:30 | 1 | 11.8 | 65.8 | 0.13 | 539 | 9.0 | 8.5 | 19.5-21.9 | 99.3 |
| 10 | 05/08/2014 18:45 | 06/08/2014 06:13 | 1 | 11.5 | 40.0 | 0.11 | 227 | 11.1 | 9.8 | 16.0-20.3 | 99.3 |
| 11 | 09/08/2014 07:41 | 09/08/2014 09:32 | 0 | 1.8 | 17.4 | 0.06 | 261 | 7.9 | 7.7 | 13.7-14.0 | 100.0 |
| 12 | 11/08/2014 20:42 | 11/08/2014 21:09 | 0 | 0.4 | 173.3 | 0.06 | 392 | 8.3 | 7.7 | 17.6-17.9 | 99.7 |
| 13 | 12/08/2014 23:04 | 13/08/2014 03:55 | 2 | 4.8 | 66.1 | 0.19 | 536 | 9.4 | 9.1 | 13.8-16.9 | 99.0 |
| 14 | 13/08/2014 18:58 | 14/08/2014 06:22 | 3 | 11.4 | 34.5 | 0.19 | 312 | 10.9 | 9.7 | 13.5-15.9 | 98.4 |
| 15 | 14/08/2014 17:35 | 14/08/2014 19:52 | 0 | 2.3 | 94.6 | 0.02 | 104 | 7.2 | 6.5 | 15.7-17.7 | 98.8 |
| 16 | 15/08/2014 18:52 | 16/08/2014 05:59 | 0 | 11.1 | 66.4 | 0.04 | 283 | 6.9 | 6.5 | 15.0-17.6 | 99.2 |
| 17 | 16/08/2014 19:45 | 17/08/2014 05:10 | 0 | 9.4 | 93.9 | 0.03 | 157 | 8.3 | 7.3 | 15.5-18.2 | 98.4 |
| 18 | 17/08/2014 10:02 | 17/08/2014 11:13 | 1 | 1.2 | 63.5 | 0.39 | 722 | 11.7 | 10.6 | 14.9-17.0 | 99.2 |
| 19 | 17/08/2014 21:57 | 18/08/2014 01:23 | 1 | 3.4 | 52.5 | 0.10 | 366 | 8.5 | 8.3 | 14.3-15.2 | 99.1 |
| 20 | 18/08/2014 08:42 | 18/08/2014 11:05 | 0 | 2.4 | --- | 0.03 | 118 | 7.2 | 6.8 | 15.0-16.5 | 98.4 |
| 21 | 21/08/2014 20:00 | 22/08/2014 13:48 | 0 | 17.8 | 57.9 | 0.02 | 109 | 7.0 | 6.5 | 15.9-20.7 | 96.3 |
| 22 | [b]23/08/2014 01:30 | 23/08/2014 09:20 | 3 | 7.8 | 43.0 | 0.21 | 624 | 9.6 | 9.4 | 16.2-17.4 | 99.6 |
| 23 | 23/08/2014 18:12 | 23/08/2014 19:54 | 0 | 1.7 | 70.6 | 0.01 | 88 | 6.8 | 6.3 | 16.8-17.8 | 99.5 |
| 24 | 25/08/2014 02:25 | 25/08/2014 06:40 | 0 | 4.2 | 29.4 | 0.01 | 79 | 5.7 | 5.3 | 13.8-15.0 | 97.8 |

[a] the arithmetic mean value

[b] the selected four typical cloud events according to the average $PM_{2.5}$ level for 28/07/2014 22:40 to 29/07/2014 04:00 (event A, 81.6 $\mu g\ m^{-3}$), 23/08/2014 01:30 to 23/08/2014 09:20 (event B, 43.0 $\mu g\ m^{-3}$), 30/07/2014 20:20 to 30/07/2014 22:40 (event C, 25.0 $\mu g\ m^{-3}$) and 25/07/2014 12:00 to 25/07/2014 21:40 (event D, 11.1 $\mu g\ m^{-3}$).



**Figure 1:**

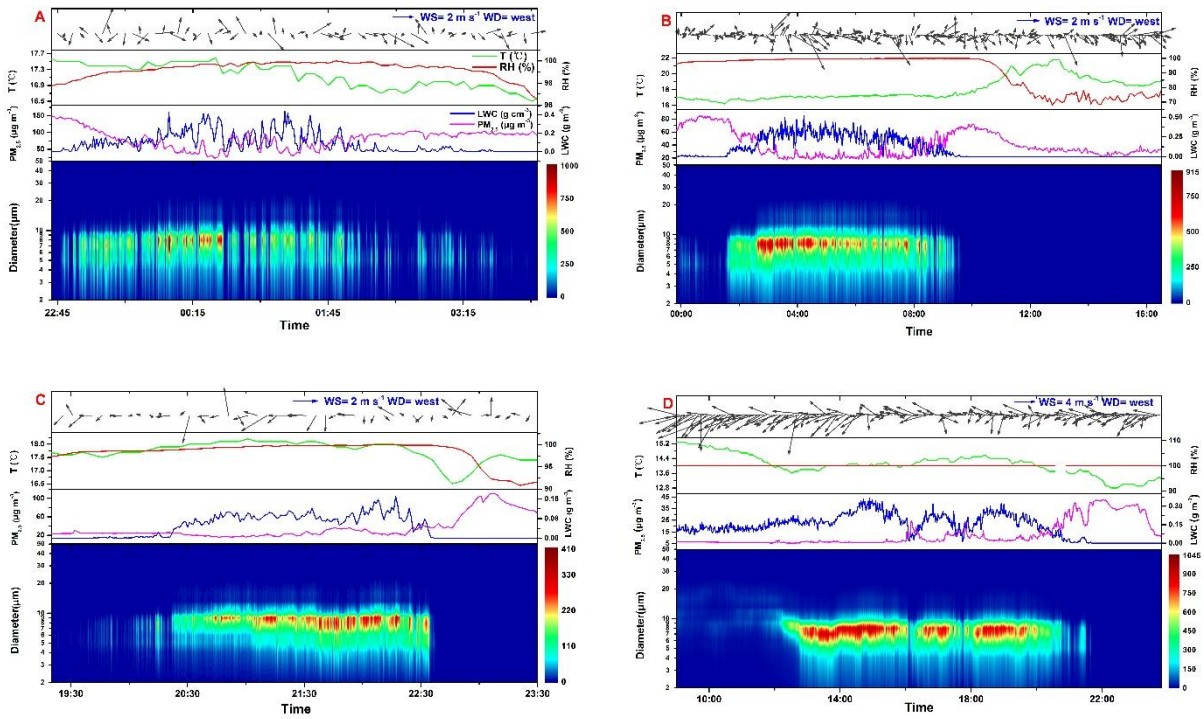

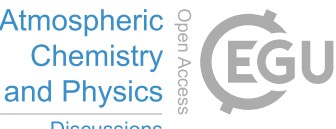



**Figure 2:**

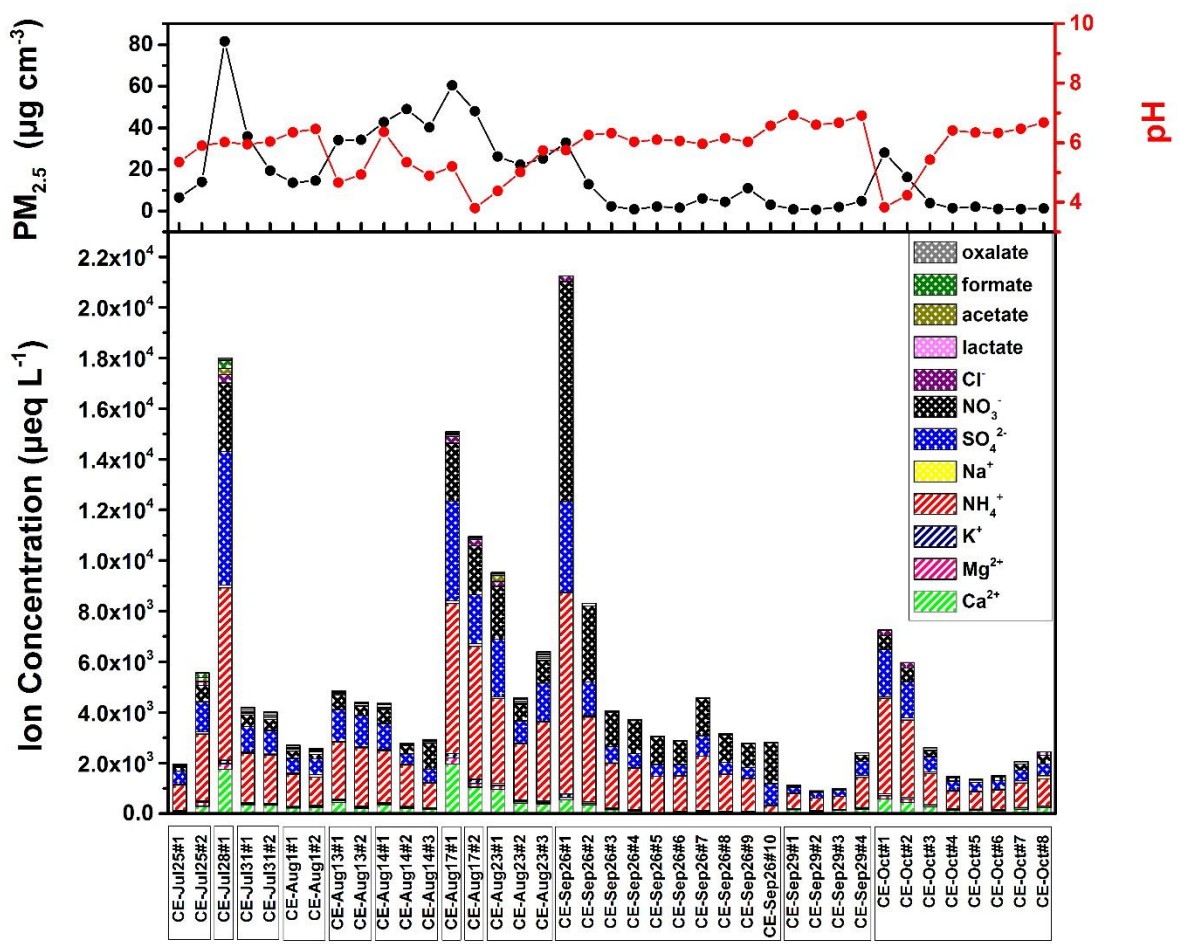





**Figure 3:**

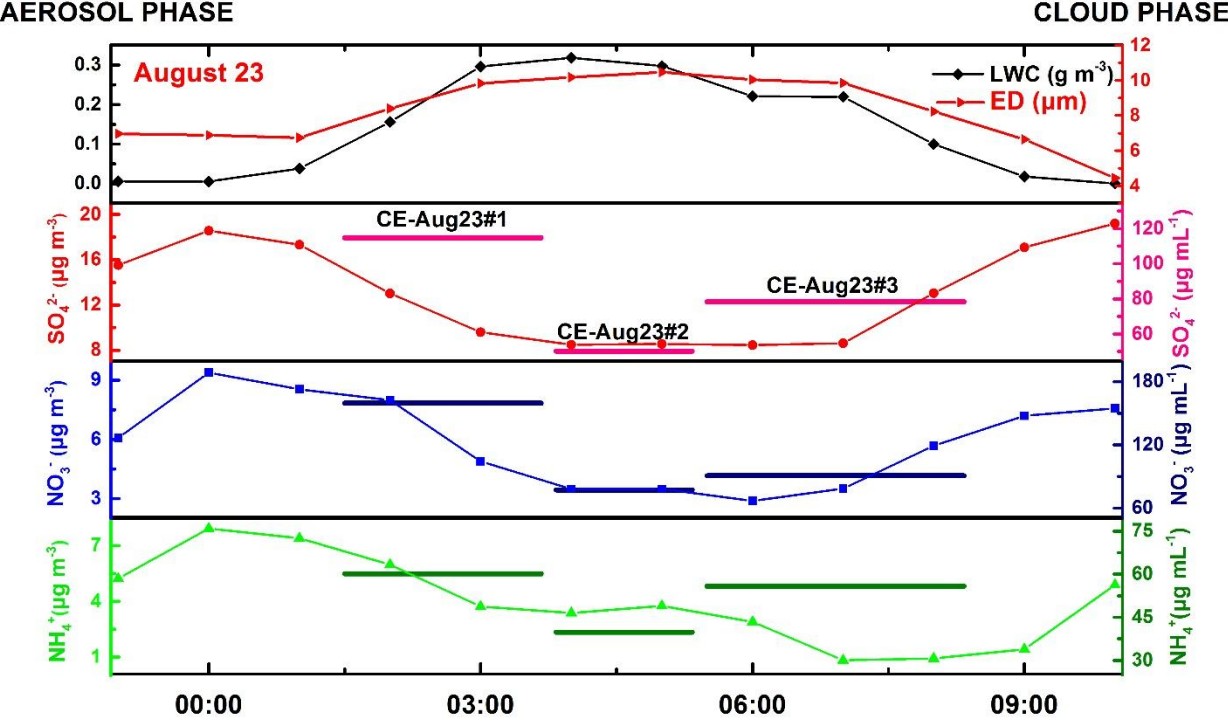





**Figure 4:**

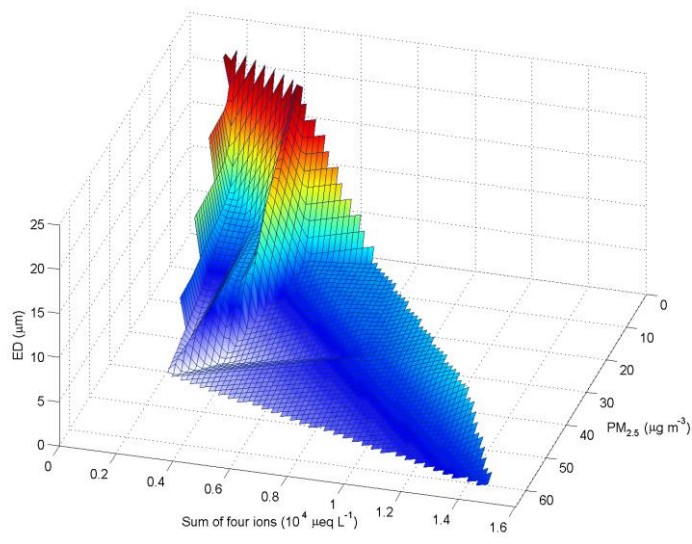

**(a)**

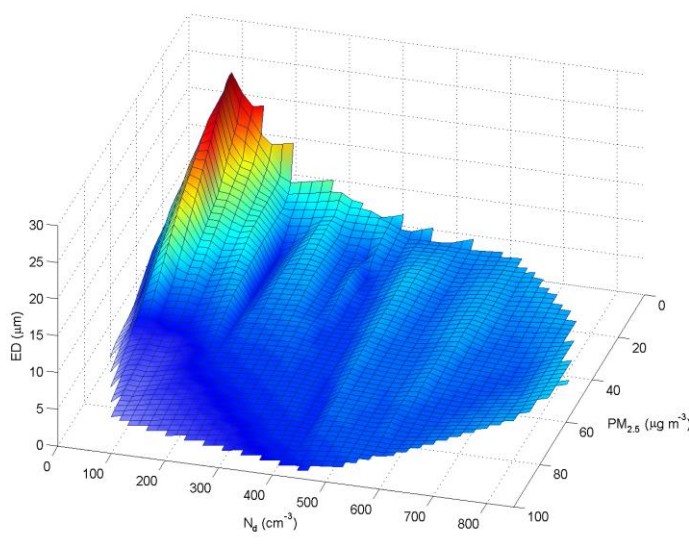

**(b)**





**Figure 5:**

**a. High PM$_{2.5}$ level**

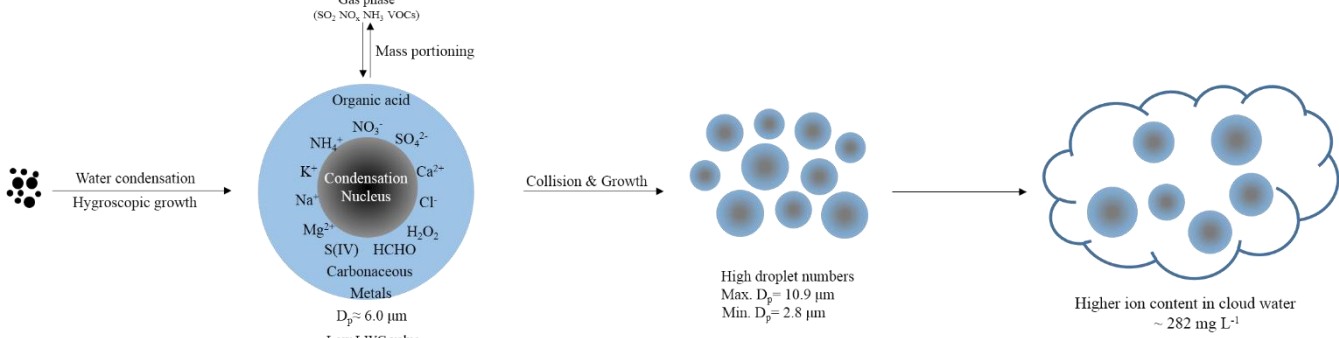

**b. Low PM$_{2.5}$ level**

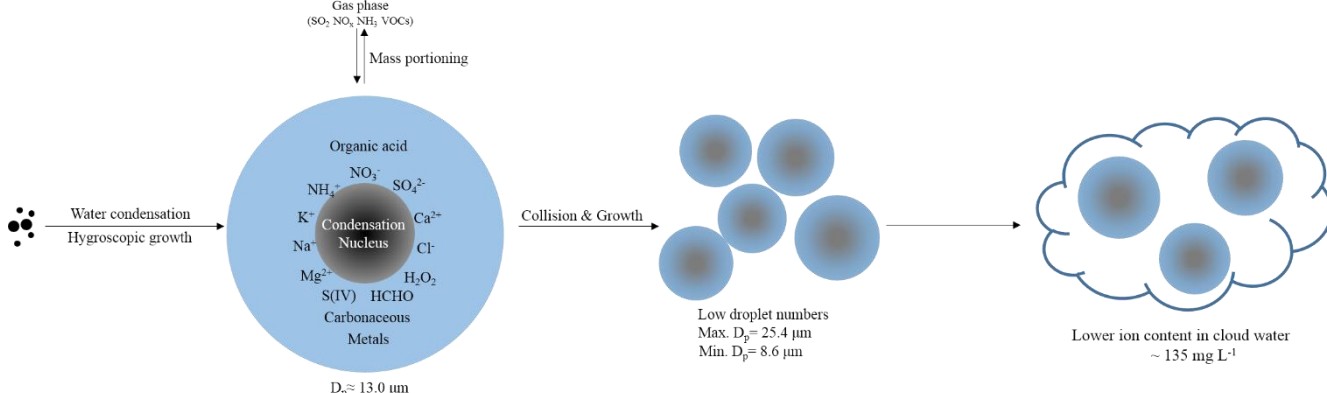