# Peer review of "Chemical composition and droplet size distribution of cloud at the summit of Mount Tai, China"

_Atmospheric Chemistry and Physics, 2016_

## Referee Comment (RC1) · Anonymous Referee #2 · 17 Mar 2017

General comments

The manuscript "Chemical composition and droplet size distribution of cloud at the summit of Mount Tai, China" presents the measurement of chemical composition including pH and soluble ions of cloud water and physical properties of cloud droplet (number, size and liquid water content) in a mountain site for cloud events strongly influenced by anthropogenic emissions. It further investigates how the chemical composition and physical properties of clouds are influenced by aerosols (using $PM_{2.5}$ mass concentration a proxy) and how clouds affect aerosol concentrations. The authors found that the pH value was higher than 2007-2008 reported in a previous study for the same site, which was attributed to the increase of $NH_4^+$ and $Ca^{2+}$. The authors further found that when $PM_{2.5}$ was higher, the concentrations of soluble ions were higher and cloud droplets were smaller. Overall, the findings of this study are interesting and this study works an evaluable case study on the interaction of aerosol with clouds, especially for the cloud with strong influence of anthropogenic emissions. While the manuscript is mostly well written and fits the scope of ACP, I have some comments before it is published on ACP. These comments are mainly meant to clarify some discussions and improve the readability.

1. In the discussion of the interaction of aerosols with clouds, it is mainly the number of CCN that affects cloud microphysical properties, not the $PM_{2.5}$ mass concentration. Although the $PM_{2.5}$ mass concentration and CCN number concentration may correlate during the cloud events studied here, it is not necessarily true in many cases because particles contributing mostly to CCN number and $PM_{2.5}$ mass concentrations may differ in size ranges, depending on the particle size distribution. In this manuscript, $PM_{2.5}$ mass concentration was used a somewhat proxy for CCN. While the particle size distribution data are not available here, the authors need at minimum discuss the limit of using $PM_{2.5}$ here.

2. Some of the discussion or statement are not quantitative enough and need further clarification or supporting data.
   For example, in Pg 6 lines 2-11 on the relationship between PM2.5 level and LWC and cloud droplet size, instead of selecting a few cloud events with higher PM2.5 and qualitatively comparing the droplets sizes in these events, a quantitative way would be plot the droplet size versus PM2.5 level. Same principle applies for the effect of RH on droplet size explanation (low RH suppressing cloud droplets size). Since the effect of PM2.5 on droplet size is anyway discussed in Sect. 3.4, the authors could consider to merge this paragraph with the discussion of Sect. 3.4.
   Pg 6, lines 21-22, "…broadened the droplet size spectra…", it would be helpful to provide the standard deviation or geometric standard deviation of the droplet size distribution, because such broadening is not clear from Fig. 1 (the green color becoming wider does not necessarily mean broadening, which could be only due to increasing concentrations in all sizes).
   Pg 7, lines 16-19, about the origins of air mass, it would be helpful to add the information on this, such as back trajectory. And the wind directions in Fig. 1 did not show consistent directions except for panel D, in which winds are mainly from eastern sector.
   Pg 7, lines 22, "the TDIC was strongly correlated with the levels of PM2.5 and cloud acidity", it looks like to me that the correlation of TDIC with acidity is not that strong if TDIC were plotted agaist pH. Unlike TDIC, pH should not only depend on the PM2.5 concentration but

also chemical compositions of PM2.5, for example, whether there are more acidic or basic compounds.

Pg 8, lines 16-19, "the increase in the concentration of NH4+ from CE-Aug23#2 to CE-Aug23#3 was much higher than those of SO42- and NO3-," it would be helpful to provide the number of increase of $NH_4^+$, $NO_3^-$, and $SO_4^{2-}$. (Do you mean the molar concentrations here?)

3. In some discussion, not enough background information is available to understand the discussion. For example, in Pg 6, line 2 "High PM2.5 levels can lead to low LWC values, which can diminish the size of the cloud droplets", at this point, I had difficulty to understand this statement here without further explanation, for exampling, using the findings from literature. Also lines 9-10, "If the RH remains constant, each CCN shares less water vapor, which leads to lower LWC values and hinders the growth of cloud droplets.", I also had difficult time understanding why it is so.

Specific comments

1. Pg 2, line 10 "…more than 30% of the total annual sulfur depositionwas deposited as a result of cloud events (Shimadera et al., 2011).", for me, that does not seem to be relevant to the arguments before on the role of non-precipitation vs. precipitation clouds.
2. Pg 4, line 3, what is effective diameter exactly defined?
3. Pg 5, line 24, "…may be attributable to the increasing consumption of agricultural fertilization and soil acidification…", I suppose this only refers to $NH_4^+$ not $Ca^{2+}$. If so, please clarify.
4. Pg 5, line 32, "This diversity was a result of the characteristic formation…", the meaning of "characteristic formation" is vague.
5. Pg 6, lines 29-30, "It should be emphasized that although the levels of PM2.5 decreased from event A to event D, there were no significant changes in the CDSD properties." What does "CDSD properties" mean? I suppose the droplet size (ED) is also a CDSD property. If so, it is affected by PM2.5 level as discussed in Sect. 3.4 and Fig. 4 and would contradict the statement here.
6. Pg 7, Sect 3.2.3, why do the two types of cloud behave differently? Because of the origin of air mass?
7. Pg 7, line 27, "CCN, especially particulate matters, are likely to be the main source of ions and acid-causing components in cloud water." I suggest to omit "especially particulate matters" because it seems to indicate that some CCNs are not particulate matters.
8. Pg 7, line 28, "… the transmission and variation…"
9. I suggest authors to further polish the languages.
10. Fig.1d, the RH is flat. Is it constantly at 100%?
11. Fig. 5, how are the values of size Dp (Dp=6.0 for high PM2.5 level and Dp=13.0 for low PM2.5 level and so on) and ion content obtained? Please clarify.

Technical comments

1. Pg 2 line 6 "…taking place multiphase chemical reactions", maybe "…multiphase chemical reactions taking place" is better.

2. Pg 3, line 17, add comma after "conductivity" and "formaldehyde" (and omit the "and" after).
3. Pg 7, line 27, "…acid-causing components…" does not sound the right wording. Please re-phrase.
4. Pg 7, line 28, "… the transmission and variation…", "transmission" does sound right, maybe "partitioning" or "exchange".
5. Pg 8, line 7, "The above results demonstrate that cloud water is a dominant sink", by "dominant" I guess that authors meant important since they did not compare with other sinks.
6. Pg 8, lines 11-14, the author emphasize the importance of dilution effect of cloud water. However, based on strong correlation of PM2.5 and TDIC regardless of the LWC level, does the correlation suggest that the dilution effect throughout all these cloud events are similar and therefore not crucial?
7. Pg 8 line 21, "….were the main hygroscopic compounds.",  add "hygroscopic compounds' of what? PM2.5?
8. Pg 8 line 31, "… Nd varied with the same PM2.5 level", change "with" to "at".

---

## Referee Comment (RC2) · Anonymous Referee #1 · 14 May 2017

Comments on "Chemical composition and droplet size distribution of cloud at the summit of Mount Tai, China" by Li et al. Clouds can affect the earth's radiation budget and regional and global climate, which are are influenced by the chemical compositions of cloud waters. However, limited studies have been conducted on the interactions between aerosols and the chemical and microphysical properties of clouds, especially in East China. Here, Li et al. report the chemical composition of 39 cloud samples that were collected at the summit of Mt. Tai in the North China Plain from July to October 2014. In addition, microphysical properties of cloud droplets, including cloud droplet size distribution (CDSD), liquid water content (LWC), and droplet number concentration were investigated. Overall, the manuscript is well written and easy to follow. I suggest it to be accepted for publication in ACP after some modifications as listed below: 1. Page 3, line 29: Any reference on the measurement of these organic acids using IC?

Please provide it. 2. Page 3, line 30-31: it's not clear to readers how OC and EC in cloud samples were measured using a sunset OC/EC analyzer. Detailed information is needed here. 3. The authors report organic acids in cloud samples from the summit of Mt. Tai. What's the main sources of the measured organic acids such as lactic and oxalic acids in cloud waters? Detailed discussion on such a point is needed. 4. Are there any correlations between organic acids and water-soluble cations in the cloud samples?

———————————————

---

## Author Comment (AC1) · 11 Jun 2017

Responds to the reviewer's comments:

We sincerely thank the reviewer for the valuable comments and suggestions concerning our manuscript entitled "Chemical composition and droplet size distribution of cloud at the summit of Mount Tai, China". These comments are all valuable and helpful for revising and improving our paper. The responses to reviewers are in blue. The changes are marked in red in the revised manuscript.

**Reviewer 2**

**General comments**
**Comment 1:**
1. In the discussion of the interaction of aerosols with clouds, it is mainly the number of CCN that affects cloud microphysical properties, not the PM2.5 mass concentration. Although the PM2.5 mass concentration and CCN number concentration may correlate during the cloud events studied here, it is not necessarily true in many cases because particles contributing mostly to CCN number and PM2.5 mass concentrations may differ in size ranges, depending on the particle size distribution. In this manuscript, PM2.5 mass concentration was used a somewhat proxy for CCN. While the particle size distribution data are not available here, the authors need at minimum discuss the limit of using PM2.5 here.
**Response:** We sincerely thanks you for your pertinent comments and valuable suggestions. We added the reason for using $PM_{2.5}$ mass concentration represent CCN in this study as in Page 7 Line 23- Line 27:

*"But for both types of cloud events, the $N_d$ significantly decreased and the $PM_{2.5}$ levels evidently increased as cloud events began to dissipate. It may due to the evaporation of water contents that condensed on the particles, which freed the CCN and formed haze. This confirmed that $PM_{2.5}$ was one of the important types of could condensation nuclei at Mt. Tai. So, $PM_{2.5}$ mass concentration was used as a proxy for CCN number concentration in this study."*

**Comment 2:**
2. Some of the discussion or statement are not quantitative enough and need further clarification or supporting data.
For example, in Pg 6 lines 2-11 on the relationship between PM2.5 level and LWC and cloud droplet size, instead of selecting a few cloud events with higher PM2.5 and qualitatively comparing the droplets sizes in these events, a quantitative way would be plot the droplet size versus PM2.5 level. Same principle applies for the effect of RH on droplet size explanation (low RH suppressing cloud droplets size). Since the effect of PM2.5 on droplet size is anyway discussed in Sect. 3.4, the authors could consider to merge this paragraph with the discussion of Sect. 3.4.
**Response:** We merged this paragraph with the discussion of Sect. 3.4.

Pg 6, lines 21-22, "…broadened the droplet size spectra…", it would be helpful to provide the

standard deviation or geometric standard deviation of the droplet size distribution, because such broadening is not clear from Fig. 1 (the green color becoming wider does not necessarily mean broadening, which could be only due to increasing concentrations in all sizes).

**Response:** Thanks for your suggestion. We added the standard deviation (SD) of cloud droplet size distribution in Fig. 1. As can be seen, both LWC and SD increased with the development of the cloud events. It represented that high LWC values could broaden the droplet size spectra. We updated Fig. 1 and added the corresponding discussion as in Page 7 Line 10- Line 12:

*"With the development of the cloud event, the standard deviation of CDSD represented a positive correlation with LWC values. It represented that high LWC could broaden the droplet size spectra and increase the range of cloud droplets."*

Pg 7, lines 16-19, about the origins of air mass, it would be helpful to add the information on this, such as back trajectory. And the wind directions in Fig. 1 did not show consistent directions except for panel D, in which winds are mainly from eastern sector.

**Response:** We used the backward trajectory analysis, but no big difference existed in the origins of the air masses of event A, event B and event C as shown in Figure R1. By re-examining the data, we considered the ambient concentrations of $PM_{2.5}$ at the beginning stage of cloud events was probably the main reason to explain the difference between type I and type II. Because the original concentrations of $PM_{2.5}$ in event A and event B were higher than those in event C and event D. We revised the discussion as in Page 7 Line 22- Line 27:

*"In type II cloud processes, the levels of $PM_{2.5}$ were relatively low at the initial stage. But for both types of cloud events, the $N_d$ significantly decreased and the $PM_{2.5}$ levels evidently increased as cloud events began to dissipate. It may due to the evaporation of water contents that condensed on the particles, which freed the CCN and formed haze. This confirmed that $PM_{2.5}$ was one of the important types of could condensation nuclei at Mt. Tai. In this study, $PM_{2.5}$ mass concentration was used as a proxy for CCN number concentration."*

[Figure]

Figure R1. Backward trajectories of air masses for four cloud events at Mt. Tai

Pg 7, lines 22, "the TDIC was strongly correlated with the levels of PM2.5 and cloud acidity", it looks like to me that the correlation of TDIC with acidity is not that strong if TDIC were plotted agaist pH. Unlike TDIC, pH should not only depend on the PM2.5 concentration but also chemical compositions of PM2.5, for example, whether there are more acidic or basic compounds.

**Response:** Here, we want to emphasize the influence of $PM_{2.5}$ on TDIC and pH values. Even though the pH of cloud samples was not only depend on the $PM_{2.5}$ concentration, the lower pH values were likely to occur at higher concentrations of $PM_{2.5}$ as can be seen in Fig. 1. We modified the description of this paragraph as in Page 7 Line 29- Page 8 Line 5:

*"As illustrated in Fig. 2, the TDIC was strongly correlated with the levels of $PM_{2.5}$. High levels of $PM_{2.5}$ normally lead to high TDIC, whereas low levels of $PM_{2.5}$ usually lead to low TDIC. The pH values of cloud samples were somewhat affected by the concentrations of $PM_{2.5}$. The lower pH values were likely to occur at higher concentrations of $PM_{2.5}$. Generally, changes of the solute concentrations in cloud water can be caused by a combination of factors such as the microphysical conditions, the CCN properties, the chemical reactions in the cloud droplets and the gas-liquid phase equilibrium (Van Pinxteren et al., 2015). Our data emphasized the crucial effect of $PM_{2.5}$ on the changes of ion concentrations. $PM_{2.5}$ are likely to be the main source of ions in cloud water."*

Pg 8, lines 16-19, "the increase in the concentration of NH4+ from CE-Aug23#2 to CEAug23#3 was much higher than those of SO42- and NO3-," it would be helpful to provide the number of increase of NH4+, NO3-, and SO42-. (Do you mean the molar concentrations here?)

**Response:** Thanks for your comments. The increase factors of $SO_4^{2-}$, $NO_3^-$ and $NH_4^+$ from CE-

Aug23#2 to CE-Aug23#3 had been described in Page 8 Line 14- Line 16. In the manuscript, we used the mass concentrations ($\mu g\ mL^{-1}$) to calculate the increase factors of ions. We revised the discussion as in Page 8 Line 26-29:

*"It should be noted that, compared with $SO_4^{2-}$ and $NO_3^-$, the concentration of $NH_4^+$ in aerosol phase did not directly increase at the dissipation stage of the cloud event. This was primarily due to the high solubility of $NH_3$, which dissolved in the cloud water and gave rise to the increase in the concentration of $NH_4^+$ in cloud sample."*

**Comment 3:**

3. In some discussion, not enough background information is available to understand the discussion. For example, in Pg 6, line 2 "High $PM_{2.5}$ levels can lead to low LWC values, which can diminish the size of the cloud droplets", at this point, I had difficulty to understand this statement here without further explanation, for exampling, using the findings from literature.
**Response:** This statement was summarized from our data in Table 3. Ackerman and colleagues (Ackerman et al., 2004) also found $PM_{2.5}$ levels could affected the LWC values and the sizes of cloud droplets. But considering the comment 2 in general comments, we merged this paragraph with the discussion of Sect. 3.4.

Also lines 9-10, "If the RH remains constant, each CCN shares less water vapor, which leads to lower LWC values and hinders the growth of cloud droplets.", I also had difficult time understanding why it is so.
**Response:** High levels of $PM_{2.5}$ can lead to a large source of CCN and intensify the competition for the ambient water vapor. The low RH could not provide sufficient water vapor, which would reduce the hygroscopic growth of aerosols and hinder the activation of droplets (Gonser et al., 2012; Liu et al., 2011). The combination of the two factors determined the low LWC and ED values of the cloud droplets. But considering the comment 2 in general comments, we merged this paragraph with the discussion of Sect. 3.4.

**Specific comments**
**Comment 1:**
Pg 2, line 10 "…more than 30% of the total annual sulfur deposition was deposited as a result of cloud events (Shimadera et al., 2011).", for me, that does not seem to be relevant to the arguments before on the role of non-precipitation vs. precipitation clouds.
**Response:** We revised the statement and updated the reference as in Page 2 Line 12-14:

*"For example, Sun and colleagues (Sun et al., 2010) found that the concentrations of ammonium, sulfate and nitrate in cloud water were at least 5.17 times higher than those in rainwater."*

**Comment 2:**
Pg 4, line 3, what is effective diameter exactly defined?
**Response:** As described in the instruction manual of Fog Monitor, the definition of ED (Effective Diameter) is the ratio of liquid water content to the optical cross sectional area of

droplets. Effective diameter in μm is calculated according to the following definition:

$$ED = \frac{3LWC}{4G\rho_w} \times 2$$

where: LWC = Liquid Water Content in g m$^{-3}$

G = The geometric cross-sectional area of water drops ($\mu m^2$) per unit volume ($\mu m^3$)

$\rho_w$ = The density of water (equal to $10^6$ g m$^{-3}$)

We added the reference in the revised manuscript as in Page 4 Line 13.

**Comment 3:**

Pg 5, line 24, "…may be attributable to the increasing consumption of agricultural fertilization and soil acidification…", I suppose this only refers to $NH_4^+$ not $Ca^{2+}$. If so, please clarify.

**Response:** Thanks for your comments. We have clarified as in Page 5 Line 31- Page 6 Line 2:

*"Especially $NH_4^+$, the VWM concentrations of $NH_4^+$ increased from 2007–2008 by factors of 1.56 (Guo et al., 2012).This may be attributable to the increasing consumption of agricultural fertilization and soil acidification (Cai et al., 2015; Xu et al., 2015)."*

**Comment 4:**

Pg 5, line 32, "This diversity was a result of the characteristic formation…", the meaning of "characteristic formation" is vague.

**Response:** We have detailedly described the "characteristic formation" as in Page 6 Line 22-Line 24:

*"Orographic cloud is a highly heterogeneous system consisting of randomly distributed air volumes with different characteristics(Gonser et al., 2012). This feature of orographic cloud generally determines the large differences in CDSD, LWC and aerosol composition of different cloud events."*

**Comment 5:**

Pg 6, lines 29-30, "It should be emphasized that although the levels of $PM_{2.5}$ decreased from event A to event D, there were no significant changes in the CDSD properties." What does "CDSD properties" mean? I suppose the droplet size (ED) is also a CDSD property. If so, it is affected by $PM_{2.5}$ level as discussed in Sect. 3.4 and Fig. 4 and would contradict the statement here.

**Response:** The "CDSD properties" here just means the size distribution obtained in Figure 1. This had been described in the first paragraph in Sect. 3.2.2 that the monitored cloud droplets were all smaller than 26 μm, mainly distributed in 6.0-9.0 μm and concentrated in 7.0 μm. In order to avoid misunderstanding, we deleted this sentence in the revised manuscript.

**Comment 6:**

Pg 7, Sect 3.2.3, why do the two types of cloud behave differently? Because of the origin of air mass?

**Response:** We applied the backward trajectory analysis, but no big difference existed in the origins of the air masses of event A, event B and event C. We considered the ambient

concentrations of PM$_{2.5}$ at the beginning stage of cloud events was probably the main reason to explain the difference between type I and type II.

**Comment 7:**

Pg 7, line 27, "CCN, especially particulate matters, are likely to be the main source of ions and acid-causing components in cloud water." I suggest to omit "especially particulate matters" because it seems to indicate that some CCNs are not particulate matters.

**Response:** Thanks for your comments. We deleted "especially particulate matters".

**Comment 8:**

Pg 7, line 28, "… the transmission and variation…"

**Response:** We changed the "transmission" as "exchange" in the paper.

**Comment 9:**

I suggest authors to further polish the languages.

**Response:** We have polished the language, and improved the grammar and expressions.

**Comment 10:**

Fig.1d, the RH is flat. Is it constantly at 100%?

**Response:** Yes. Our data was constantly at 100% during this period. In 2015, we also carried out the cloud observation experiment at Mt.Tai. And we found the same phenomenon such as on June 29$^{th}$ and on July 29$^{th}$. The RH remained at 100% for a long time.

[Figure]

[Figure]

**Comment 11:**

Fig. 5, how are the values of size $D_p$ ($D_p$=6.0 for high $PM_{2.5}$ level and $D_p$=13.0 for low $PM_{2.5}$ level and so on) and ion content obtained? Please clarify.

**Response:** The cloud samples were divided into "high $PM_{2.5}$ level" and "low $PM_{2.5}$ level". The data used for obtaining the values in Fig. 5 was shown in Table R1. The values of $D_p$ and the concentrations of ion contents in Fig. 5 were obtained by calculating the average values of cloud samples in "high $PM_{2.5}$ level" and "low $PM_{2.5}$ level". In the previous version, we chose 40 μg m$^{-3}$ of $PM_{2.5}$ as the standard to distinguish the "high $PM_{2.5}$ level" and the "low $PM_{2.5}$ level". Through comparing the air quality standard in China and in America, we considered that using 35 μg m$^{-3}$ of $PM_{2.5}$ as the standard to divide the cloud samples as the "high $PM_{2.5}$ level" and the "low $PM_{2.5}$ level" was better. It should be emphasized that the change of the standard didn't alter the mechanism of cloud droplet formation found in this paper. The clarification was added in the revised manuscript as in Page 10 Line 1- Line 3:

*"According to the concentrations of $PM_{2.5}$, cloud events were divided into two categories. One was the $PM_{2.5}$ concentrations greater than 35 μg m$^{-3}$. The other was the $PM_{2.5}$ concentrations less than or equal to 35 μg m$^{-3}$."*

Table R1. Data of 17 cloud samples for calculating $D_p$ and ion contents in Fig. 5

| Average ED* µm | Average Ion Contents* mg L⁻¹ | No. | Cloud Sample | PM₂.₅ µg m⁻³ | Ion Contents mg L⁻¹ | ED µm |
|---|---|---|---|---|---|---|
| low PM₂.₅ level (PM₂.₅ ≤ 35 µg m⁻³) 13.5 | 136.9 | 1 | CE-Jul25#1 | 6.51 | 53.72 | 25.47 |
| | | 2 | CE-Aug1#1 | 13.63 | 72.72 | 11.60 |
| | | 3 | CE-Jul25#2 | 14.04 | 150.01 | 25.47 |
| | | 4 | CE-Aug1#2 | 14.59 | 66.17 | 14.57 |
| | | 5 | CE-Jul31#2 | 19.44 | 112.99 | 9.70 |
| | | 6 | CE-Aug23#2 | 22.28 | 132.04 | 10.35 |
| | | 7 | CE-Aug23#3 | 25.20 | 191.05 | 9.07 |
| | | 8 | CE-Aug23#1 | 26.22 | 317.56 | 9.73 |
| | | 9 | CE-Aug13#1 | 34.11 | 145.54 | 10.54 |
| | | 10 | CE-Aug13#2 | 34.22 | 127.07 | 8.60 |
| high PM₂.₅ level (PM₂.₅ > 35 µg m⁻³) 6.5 | 258.9 | 1 | CE-Jul31#1 | 35.91 | 118.69 | 9.09 |
| | | 2 | CE-Aug14#3 | 40.23 | 109.94 | 3.47 |
| | | 3 | CE-Aug14#1 | 42.74 | 127.47 | 2.75 |
| | | 4 | CE-Aug17#2 | 47.98 | 328.07 | 8.39 |
| | | 5 | CE-Aug14#2 | 48.98 | 73.76 | 3.01 |
| | | 6 | CE-Aug17#1 | 60.38 | 476.38 | 10.89 |
| | | 7 | CE-Jul28#1 | 81.59 | 578.15 | 7.71 |

\* Average ED represented "$D_p$" in Fig. 5

\* Average Ion Contents represented "ion content" in Fig. 5

We have revised the Fig. 5 as following:

[Figure]

**Technical comments**

**Comment 1:**

Pg 2 line 6 "…taking place multiphase chemical reactions", maybe "…multiphase chemical reactions taking place" is better.

**Response:** Thanks for your comments. We revised "…taking place multiphase chemical reactions" as "…multiphase chemical reactions taking place" in the revised manuscript.

**Comment 2:**

Pg 3, line 17, add comma after "conductivity" and "formaldehyde" (and omit the "and" after).

**Response:** We rewrote the sentence as in Page 3 Line 22-23:

*"The pH, the electrical conductivity, the concentrations of sulfur(IV), formaldehyde, hydrogen peroxide were measured immediately after sampling."*

**Comment 3:**

Pg 7, line 27, "…acid-causing components…" does not sound the right wording. Please rephrase.

**Response:** Thanks for your comments. Maybe "acidic compounds" was more appropriate.

**Comment 4:**

Pg 7, line 28, "… the transmission and variation…", "transmission" does sound right, maybe "partitioning" or "exchange".

**Response:** We changed the "transmission" as "exchange" in the revised manuscript.

**Comment 5:**

Pg 8, line 7, "The above results demonstrate that cloud water is a dominant sink", by "dominant" I guess that authors meant important since they did not compare with other sinks.

**Response:** We modified "dominant" as "important" in the revised manuscript.

**Comment 6:**

Pg 8, lines 11-14, the author emphasize the importance of dilution effect of cloud water. However, based on strong correlation of $PM_{2.5}$ and TDIC regardless of the LWC level, does the correlation suggest that the dilution effect throughout all these cloud events are similar and therefore not crucial?

**Response:** The dilution effect of LWC on ion concentrations could not be ignored. Previous studies also found the inverse relationship between total ionic content and LWC in cloud samples (Aleksic and Dukett, 2010; Elbert et al., 2000). Through this study, we could only confirmed that both $PM_{2.5}$ and LWC would affect the estimation of TDIC in the cloud samples. This statement may become clearer if re-written as in Page 8 Line 23-Line 25:

*"The similar variation trends of $SO_4^{2-}$, $NO_3^-$ and $NH_4^+$ in both aerosol phase and cloud phase confirmed that LWC was an important factor affecting the ion concentrations in the cloud water at Mt. Tai (Aleksic and Dukett, 2010; Elbert et al., 2000)."*

**Comment 7:**

Pg 8 line 21, "….were the main hygroscopic compounds.", add "hygroscopic compounds' of what? PM$_{2.5}$?

**Response:** Thanks for your comments. We added "…of particulate matters" as in Page 8 Line 31- Page 9 Line 2:

*"Secondary inorganic aerosols especially ammonium sulfate and ammonium nitrate were the main hygroscopic compounds of particulate matters."*

**Comment 8:**

Pg 8 line 31, "… N$_d$ varied with the same PM$_{2.5}$ level", change "with" to "at".

**Response:** Thanks for your comments. We have changed "with" to "at" as in the revised manuscript.

**References**

Ackerman, A.S., Kirkpatrick, M.P., Stevens, D.E., Toon, O.B. (2004) The impact of humidity above stratiform clouds on indirect aerosol climate forcing. Nature 432, 1014.

Aleksic, N., Dukett, J.E. (2010) Probabilistic relationship between liquid water content and ion concentrations in cloud water. Atmospheric Research 98, 400-405.

Cai, Z., Wang, B., Xu, M., Zhang, H., He, X., Zhang, L., Gao, S. (2015) Intensified soil acidification from chemical N fertilization and prevention by manure in an 18-year field experiment in the red soil of southern China. Journal of Soils & Sediments 15, 260-270.

Elbert, W., Hoffmann, M.R., Krämer, M., Schmitt, G., Andreae, M.O. (2000) Control of solute concentrations in cloud and fog water by liquid water content. Atmospheric Environment 34, 1109-1122.

Gonser, S.G., Klemm, O., Griessbaum, F., Chang, S.C., Chu, H.S., Hsia, Y.J. (2012) The Relation Between Humidity and Liquid Water Content in Fog: An Experimental Approach. Pure & Applied Geophysics 169, 1-13.

Guo, J., Wang, Y., Shen, X., Wang, Z., Lee, T., Wang, X., Li, P., Sun, M., Collett Jr, J.L., Wang, W., Wang, T. (2012) Characterization of cloud water chemistry at Mount Tai, China: Seasonal variation, anthropogenic impact, and cloud processing. Atmospheric Environment 60, 467-476.

Liu, P.F., Zhao, C.S., Bel, T.G., Hallbauer, E., Nowak, A., Ran, L., Xu, W.Y., Deng, Z.Z., Ma, N., Mildenberger, K. (2011) Hygroscopic properties of aerosol particles at high relative humidity and their diurnal variations in the North China Plain. Atmospheric Chemistry & Physics 11, 3479-3494.

Sun, M., Wang, Y., Wang, T., Fan, S., Wang, W., Li, P., Guo, J., Li, Y. (2010) Cloud and the corresponding precipitation chemistry in south China: Water‐soluble components and pollution transport. Journal of Geophysical Research: Atmospheres 115.

Van Pinxteren, D., Fomba, K.W., Mertes, S., Müller, K., Spindler, G., Schneider, J., Lee, T., Collett, J., Herrmann, H. (2015) Cloud water composition during HCCT-2010: Scavenging efficiencies, solute concentrations, and droplet size dependence of inorganic ions and dissolved organic carbon. Atmospheric Chemistry & Physics 15, 24311-24368.

Xu, P., Liao, Y.J., Lin, Y.H., Zhao, C.X., Yan, C.H., Cao, M.N., Wang, G.S., Luan, S.J. (2015) High-resolution inventory of ammonia emissions from agricultural fertilizer in China from 1978 to 2008. Atmospheric Chemistry & Physics 15, 25299-25327.

---

## Author Comment (AC2) · 11 Jun 2017

Responds to the reviewer's comments:

We sincerely thank the reviewer for the valuable comments and suggestions concerning our manuscript entitled "Chemical composition and droplet size distribution of cloud at the summit of Mount Tai, China". These comments are all valuable and helpful for revising and improving our paper. The responses to reviewers are in blue. The changes are marked in red in the revised manuscript.

**Reviewer 1**

**Comment 1:**
1. Page 3, line 29: Any reference on the measurement of these organic acids using IC? Please provide it.
**Response:** We sincerely thank you for your pertinent comments and valuable suggestions. We cited the relevant references as in Page 4 Line 4.

**Comment 2:**
2. Page 3, line 30-31: it's not clear to readers how OC and EC in cloud samples were measured using a sunset OC/EC analyzer. Detailed information is needed here.
**Response:** We added the detailed information about the measurement of OC and EC as in Page 4 Line 5- Line 9:

*"The concentrations of organic carbon (OC) and elemental carbon (EC) in cloud water were determined using a thermal-optical transmittance (TOT) carbon analyzer (Sunset Laboratory, Tigard, OR, USA). For each cloud sample, certain microliters were dropped on the surface of a small standard size punch (~1.5 $cm^2$) from a pre-combusted quartz filter and analyzed based on the NIOSH protocol 870 TOT program (Khan et al., 2009; Xu et al., 2017)."*

**Comment 3:**
3. The authors report organic acids in cloud samples from the summit of Mt. Tai. What's the main sources of the measured organic acids such as lactic and oxalic acids in cloud waters? Detailed discussion on such a point is needed.
**Response:** We have added the discussion on the sources of organic acids in cloud samples as in Page 6 Line 5- Line 16:

*"The VWM concentrations of acetate, lactate, formate and oxalate were 4.1, 3.0, 1.75 and 0.81 mg $L^{-1}$, respectively, accounting for 7.01% of TDIC. Based on the sources or source strengths of formic acid and acetic acid, the formic-to-acetic acid ratio (F/A) cloud be used as an indicator to determine the sources of organic acids (Sun et al., 2016; Tan et al., 2010). Low ratio indicated the important role of direct emissions (such as biomass emission, combustion activities and aotumobile exhaust) whereas high ratio indicated the in situ photochemical generation of formic acid (Talbot et al., 1988; Tanner and Law, 2003). In the collected cloud samples, formic acid and acetic acid were highly correlated (r=0.758, p $\leq$ 0.01). F/A was about 0.78 (lower than 1), figuring out direct emissions were important sources of organic acids*

*(Kieber et al., 2002; Li et al., 2011). Oxalic acid was significantly correlated with formic acid*
*(r=0.667, p ≤ 0.01) and acetic acid (r=0.638, p ≤ 0.01). This implied that formic acid, acetic*
*acid and oxalic acid were probably emitted from the same sources and/or accumulated under*
*similar physical conditions (Tanner and Law, 2003). No significant correlations were found*
*between lactic acid and the other three carboxylic acids. No significant correlations were found*
*between lactic acid and other water-soluble ions in the cloud samples. It implied that the*
*emission source of lactic acid was different from formic, acetic and oxalic acids."*

**Comment 4:**
4. Are there any correlations between organic acids and water-soluble cations in the cloud samples?

**Response:** The correlation coefficients between organic acids and water-soluble ions were calculated as shown in Table R1. No significant correlations were found between lactate and water-soluble cations. From acetate to oxalate, the correlations between carboxylic acids and water-soluble cations gradually increased. Especially oxalate, it was strongly correlated with all measured cations. This indicated that although formate, acetate and oxalate mainly derived from direct emissions, no characteristic sources were found for these three carboxylic acids

Table R1. The correlation coefficients between four carboxylic acids and water-soluble ions in the cloud samples.

|  | lactate | acetate | formate | oxalate |
|---|---|---|---|---|
| lactate | 1.000 | | | |
| acetate | -0.292 | 1.000 | | |
| formate | -0.215 | $0.758^{**}$ | 1.000 | |
| oxalate | -0.015 | $0.638^{**}$ | $0.667^{**}$ | 1.000 |
| $Cl^-$ | -0.106 | $0.524^*$ | $0.617^{**}$ | $0.860^{**}$ |
| $NO_3^-$ | -0.027 | $0.533^*$ | $0.484^*$ | $0.901^{**}$ |
| $SO_4^{2-}$ | -0.066 | $0.554^*$ | $0.688^{**}$ | $0.959^{**}$ |
| $nss\text{-}SO_4^{2-}$ | -0.065 | $0.554^*$ | $0.687^{**}$ | $0.959^{**}$ |
| $Na^+$ | -0.140 | 0.438 | $0.558^*$ | $0.673^{**}$ |
| $NH_4^+$ | -0.060 | 0.456 | $0.574^*$ | $0.898^{**}$ |
| $K^+$ | -0.039 | 0.289 | 0.339 | $0.793^{**}$ |
| $Mg^{2+}$ | -0.009 | 0.459 | $0.554^*$ | $0.896^{**}$ |
| $Ca^{2+}$ | 0.041 | 0.374 | 0.478 | $0.912^{**}$ |

$** \ p \leq 0.01 \ * \ p \leq 0.05$

**Reference**
Khan, A.J., Swami, K., Ahmed, T., Bari, A., Shareef, A., Husain, L. (2009) Determination of elemental carbon in lake sediments using a thermal–optical transmittance (TOT) method. Atmospheric Environment 43, 5989-5995.
Kieber, R.J., Peake, B., Willey, J.D., Avery, G.B. (2002) Dissolved organic carbon and organic acids in coastal New Zealand rainwater. Atmospheric Environment 36, 3557-3563.
Li, P., Li, X., Yang, C., Wang, X., Chen, J., Collett Jr, J.L. (2011) Fog water chemistry in Shanghai. Atmospheric Environment 45, 4034-4041.

Sun, X., Wang, Y., Li, H., Yang, X., Sun, L., Wang, X., Wang, T., Wang, W. (2016) Organic acids in cloud water and rainwater at a mountain site in acid rain areas of South China. Environmental Science and Pollution Research 23, 9529.

Talbot, R.W., Beecher, K.M., Harriss, R.C., Iii, W.R.C. (1988) Atmospheric Geochemistry of Formic and Acetic Acids at a MidLatitude Temperate Site. Journal of Geophysical Research Atmospheres 93, 1638-1652.

Tan, Y., Carlton, A.G., Seitzinger, S.P., Turpin, B.J. (2010) SOA from methylglyoxal in clouds and wet aerosols: Measurement and prediction of key products. Atmospheric Environment 44, 5218-5226.

Tanner, P.A., Law, P.T. (2003) Organic Acids in the Atmosphere and Bulk Deposition of Hong Kong. Water Air & Soil Pollution 142, 279-297.

Xu, W., Wang, F., Li, J., Tian, L., Jiang, X., Yang, J., Chen, B. (2017) Historical variation in black carbon deposition and sources to Northern China sediments. Chemosphere 172, 242-248.